# What You See is What You Get: Principled Deep Learning via Distributional Generalization

**Bogdan Kulynych**[1][*]  **Yao-Yuan Yang**[2][*]  **Yaodong Yu**[3]  **Jarosław Błasiok**[4]  **Preetum Nakkiran**[2]
[1]EPFL  [2]UC San Diego  [3]UC Berkeley  [4]Columbia University
[*] denotes equal contribution.

## Abstract

Having similar behavior at training time and test time—what we call a "What You See Is What You Get" (WYSIWYG) property—is desirable in machine learning. Models trained with standard stochastic gradient descent (SGD), however, do not necessarily have this property, as their complex behaviors such as robustness or subgroup performance can differ drastically between training and test time. In contrast, we show that Differentially-Private (DP) training provably ensures the high-level WYSIWYG property, which we quantify using a notion of distributional generalization. Applying this connection, we introduce new conceptual tools for designing deep-learning methods by reducing generalization concerns to optimization ones: to mitigate unwanted behavior at test time, it is provably sufficient to mitigate this behavior on the training data. By applying this novel design principle, which bypasses "pathologies" of SGD, we construct simple algorithms that are competitive with SOTA in several distributional-robustness applications, significantly improve the privacy vs. disparate impact trade-off of DP-SGD, and mitigate robust overfitting in adversarial training. Finally, we also improve on theoretical bounds relating DP, stability, and distributional generalization.

## 1  Introduction

Much of machine learning (ML), both in theory and in practice, operates under two assumptions. First, we have independent and identically distributed (i.i.d.) samples. Second, we care only about a single averaged scalar metric (error, loss, risk). Under these assumptions, we have mature methods and theory: Modern learning methods excel when trained on i.i.d. data to directly optimize a scalar loss, and there are many theoretical tools for reasoning about *generalization*, which explain when does optimization of a scalar on the training data translates to similar values of this scalar at test time.

The focus on scalar metrics such as average error, however, misses many theoretically, practically, and socially relevant aspects of model performance. For example, models with small *average* error often have high error on salient minority subgroups [Buolamwini and Gebru, 2018, Koenecke et al., 2020]. In general, ML models are applied to the heterogeneous and long-tailed data distributions of the real world [Van Horn and Perona, 2017]. Attempting to summarize their complex behavior with only a single scalar misses many rich and important aspects of learning.

These issues are compounded for modern overparameterized networks, as their nuanced test-time behavior is not reflected at training time. For example, consider the setting of *importance sampling*: suppose we know that a certain subgroup of inputs is underrepresented in the training data compared to the test distribution (breaking the i.i.d. assumption). For underparameterized models, we can upsample this underrepresented group to account for the distribution shift [see, e.g., Gretton et al., 2009]. This approach, however, is known to empirically fail for overparameterized models [Byrd and Lipton, 2019]. Because "what you see" (on the training data) is not "what you get" (at test time), we cannot make principled train-time interventions to affect test-time behaviors. This issue extends

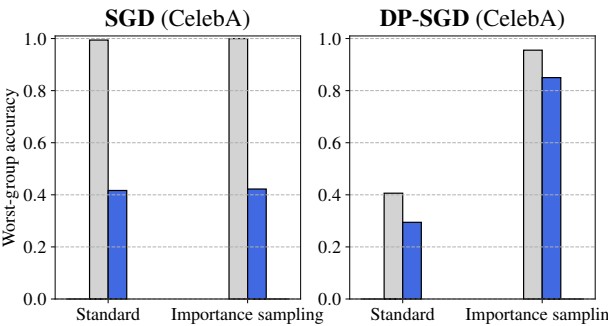

Figure 1: **Differential privacy ensures the desired behavior of importance sampling on test data.** The train and test accuracy of ResNets on CelebA, evaluated on the worst-performing ("male, blond") subgroup. *Left:* Standard SGD has a large generalization gap on this subgroup, and Importance Sampling (IS) has little effect. *Right:* DP-SGD provably has small generalization gap on all subgroups, and IS improves subgroup performance as intended. See Section 5 for details.

beyond importance sampling. For example, theoretically principled methods for distributionally robust optimization (e.g. Namkoong and Duchi [2016]) fail for overparameterized deep networks, and require ad-hoc modifications [Sagawa et al., 2019a].

We develop a theoretical framework which sheds light on these existing issues, and leads to improved practical methods in privacy, fairness, and distributional robustness. The core object in our framework is what we call the "What You See Is What You Get" (WYSIWYG) property. A training procedure with the WYSIWYG property does *not* exhibit the "pathologies" of standard stochastic gradient descent (SGD): all test-time behaviors will be expressed on the training data as well, and there will be "no surprises" in generalization.

**What You See Is What You Get (WYSIWYG) as a Design Principle.** The WYSIWYG property is desirable for two reasons. The first is diagnostic: as there are "no surprises" at test time, all properties of a model at test time are already evident at the training stage. It cannot be the case, for example, that a WYSIWYG model has small disparate impact on the training data, but large disparate impact at test time. The second reason is algorithmic: to mitigate *any* unwanted test-time behavior, it is sufficient to mitigate this behavior on the training data. This means that algorithm designers can be concerned only with achieving desirable behavior at train time, as the WYSIWYG property guarantees it holds at test time too. In practice, this enables the usage of many theoretically principled algorithms which were developed in the underparameterized regime to also apply in the modern overparameterized (deep learning) setting. For example, we find that interventions such as importance sampling, or algorithms for distributionally robust optimization, which fail without additional regularization, work exactly as intended with WYSIWYG (See Figure 1 for an illustration).

As WYSIWYG is a high-level conceptual property, we have to formalize it to use in computational practice. We do so using the notion of *Distributional Generalization* (DG), as introduced by Nakkiran and Bansal [2020], Kulynych et al. [2022]. If classical generalization ensures that the values of the model's loss on the training dataset and at test time are close on average [Shalev-Shwartz et al., 2010], distributional generalization ensures that values of any other bounded test function—not only loss—are close on training and test time. We say that a model which satisfies an appropriately high level of distributional generalization exhibits the WYSIWYG property.

**Achieving DG in Practice.** Our key observation is that distributional generalization is formally implied by *differential privacy* (DP) [Dwork et al., 2006, 2014]). The spirit of this observation is not novel: DP training is known to satisfy much stronger notions of generalization (e.g., *robust generalization*, see Section 6 for more details), and stability than standard SGD [Dwork et al., 2015a, Cummings et al., 2016, Bassily et al., 2016, Steinke and Zakynthinou, 2020]. We show that a similar connection holds for the notion of distributional generalization, and prove (and improve) tight bounds relating DP, stability, and DG. This guarantees the WYSIWYG property for any method that is differentially-private, including DP-SGD on deep neural networks [Abadi et al., 2016].

We demonstrate how DG can be a useful design principle in three concrete settings. First, we show that we can mitigate disparate impact of DP training [Bagdasaryan et al., 2019, Pujol et al., 2020] by leveraging importance sampling. Second, we study the setting of distributionally robust

optimization [e.g., Sagawa et al., 2019a, Hu et al., 2018]. We show how ideas from DP can be used to construct heuristic optimizers, which do not formally satisfy DP, yet empirically exhibit DG. Our heuristics lead to competitive results with SOTA algorithms in five datasets in the distributional robustness setting. Third, we show that the heuristic optimizer is also capable of reducing overfitting of adversarial loss in adversarial training [Madry et al., 2018, Zhang et al., 2019, Rice et al., 2020].

**Our Contributions.** We develop the theoretical connection between Differential Privacy (DP) and Distributional Generalization (DG), and we leverage our theory to improve empirical performance in privacy, fairness, and robustness applications. Theoretically (Sections 2 to 4):

1. We provide tighter bounds than previously reported connecting DP and strong forms of generalization, and show that DP training methods satisfy DG, thus the WYSIWYG property.

2. We introduce DP-IS-SGD, an importance-sampling version of DP-SGD, and show it satisfies DP and DG.

Experimentally (Section 5):

1. We use our framework to shed light on *disparate impact*: The disparity in accuracy across groups at test time are provably reflected by the accuracy disparity *on the train dataset*.

2. We use our DP-IS-SGD algorithm to largely mitigate the disparate impact of DP using importance sampling.

3. Based on our theoretical intuitions, we propose a DP-inspired heuristic: addition of gradient noise. We find this empirically achieves competitive and even improved results in several DRO settings, and reduces overfitting of adversarial loss in adversarial training.

Taken together, our results emphasize the central role of the WYSIWYG property in designing machine learning algorithms which avoid the "pathologies" of standard SGD. We also establish DP as a useful tool for achieving WYSIWYG, thus extend its applications further beyond privacy.

## 2 Theory of "What You See is What You Get" Generalization

We first review the notion of distributional generalization and demonstrate why it captures the WYSIWYG property. Second, we show that strong stability notions imply distributional generalization. Finally, we improve on the known stability guarantees of differential privacy. As a result, we extend the connections between differential privacy, stability, and generalization to *distributional* generalization, showing that stability and privacy imply the WYSIWYG property.

**Notation.** We consider a learning task with a set of examples $\mathbb{X}$ and labels $\mathbb{Y}$. We assume that a *source distribution* of labeled examples $z \triangleq (x, y) \sim \mathcal{D}$ is defined over $\mathbb{D} = \mathbb{X} \times \mathbb{Y}$. Given an i.i.d.-sampled *dataset* $S \sim \mathcal{D}^n$ of size $n$, we use a randomized *training algorithm* $\mathcal{T}(S)$ that outputs a model's parameter vector $\theta$ from the set $\Theta$. We denote by $f_\theta(x)$ the resulting prediction function.

### 2.1 Distributional Generalization and WYSIWYG

If on-average generalization [Shalev-Shwartz et al., 2010] guarantees closeness only of loss values on train and test data, distributional generalization (DG) also guarantees closeness of values of all test functions $\phi(z; \theta) \in [0, 1]$ beyond only loss:

**Definition 2.1** (Based on Nakkiran and Bansal [2020]). An algorithm $\mathcal{T}(S)$ satisfies $\delta$-distributional generalization if for all $\phi : \mathbb{D} \times \Theta \to [0, 1]$,

$$\left| \mathbb{E}_{\substack{S \sim \mathcal{D}^n \\ z \sim S}} \phi\big(z; \mathcal{T}(S)\big) - \mathbb{E}_{\substack{S \sim \mathcal{D}^n \\ z \sim \mathcal{D}}} \phi\big(z; \mathcal{T}(S)\big) \right| \leq \delta. \tag{1}$$

By the variational characterization of the total-variation (TV) distance [see, e.g., Polyanskiy and Wu, 2014, Chapter 6.3], Equation (1) is equivalent to the bound $d_{\mathsf{TV}}(P_1, P_0) \leq \delta$, where $P_1$ and $P_0$ are both distributions of $\big(z, \mathcal{T}(S)\big)$ over the randomness of $S \sim \mathcal{D}^n$ and the training algorithm $\mathcal{T}(\cdot)$, with the difference that $z \sim S$ in the case of $P_1$ (train), and $z \sim \mathcal{D}$ in the case of $P_0$ (test).

It might seem that DG only ensures average closeness of bounded tests on train and test data. This is not, however, the full picture. Consider generalization in terms of a broader class of functions:

**Definition 2.2** (Kulynych et al. [2022]). An algorithm $\mathcal{T}(S)$ satisfies $(\delta, \pi)$-distributional generalization if for a given property function $\pi : \mathbb{D} \times \Theta \to \mathbb{R}^k$ it holds that $d_{\mathsf{TV}}(\pi_\sharp P_1, \pi_\sharp P_0) \le \delta$, where $\pi_\sharp P$ is the distribution of $\pi(T)$ for $T \sim P$.

Because TV distance is preserved under post-processing, we can see that $\delta$-distributional generalization implies $(\delta, \pi)$-distributional generalization for *all* property functions. Informally, $\delta$-DG means that for *all* numeric property functions $\pi(z; \theta)$ of a model, the distributions of the property values are close on the train and test data, on average. This fact captures the high-level idea of the *"What You See is What You Get"* (WYSIWYG) guarantee. Some example property functions:

- *Subgroup loss:* $\pi(z; \theta) = \mathbb{1}\{z \in G\} \cdot \ell(z; \theta)$, for some subgroup $G \subset \mathbb{D}$.
- *Counterfactual fairness:* $\pi((x, y); \theta) = f_\theta(x') - f_\theta(x)$, where $x'$ is a counterfactual version of $x$ had it had a different value of a sensitive attribute [Kusner et al., 2017].
- *Robustness to corruptions:* $\pi(z; \theta) = \ell(A(z); \theta)$, where $A(x)$ is a possibly randomized transformation that distorts the example, e.g., by adding Gaussian noise.
- *Adversarial robustness:* $\pi(z; \theta) = \ell(A_\theta(z); \theta)$, where $A_\theta(z)$ is an adversarial example, e.g. generated using the PGD attack [Madry et al., 2018].

In the next sections, we show how a training algorithm can provably satisfy DG and therefore provide WYSIWYG guarantees for all properties, including the ones above.

## 2.2 Distributional Generalization from Stability and Differential Privacy

The connections between privacy, stability, and generalization are well-known. In particular, stability of the learning algorithm—its non-sensitivity to limited changes in the training data—implies generalization [Bousquet and Elisseeff, 2002, Shalev-Shwartz et al., 2010]. In turn, differential privacy implies strong forms of stability, thus ensuring generalization through the chain Privacy $\Rightarrow$ Stability $\Rightarrow$ Generalization [Raskhodnikova et al., 2008, Dwork et al., 2015b,a, Wang et al., 2016].

Let us formally define differential privacy:

**Definition 2.3** (Differential Privacy [Dwork et al., 2006, 2014]). An algorithm $\mathcal{T}(S)$ is $(\epsilon, \delta)$-differentially private (DP) if for any two *neighbouring datasets*—differing by one example—$S$, $S'$ of size $n$, for any subset $K \subseteq \Theta$ it holds that $\Pr[\mathcal{T}(S) \in K] \le \exp(\epsilon)\Pr[\mathcal{T}(S') \in K] + \delta$.

DP mathematically encodes a notion of plausible deniability of the inclusion of an example in the dataset. However, it can also be thought as a strong form of stability [Dwork et al., 2015b]. As such, DP implies other notions of stability. We consider the following notion, which has been studied in the literature under multiple names. In the context of privacy, it is equivalent to $(0, \delta)$-differential privacy, and has been called additive differential privacy [Geng et al., 2019], and total-variation privacy [Barber and Duchi, 2014]. In the context of learning, it has been called total-variation (TV) stability [Bassily et al., 2016]. We take this last approach and refer to it as TV stability:

**Definition 2.4** (TV Stability). An algorithm $\mathcal{T}(S)$ is $\delta$-TV stable if for any two *neighbouring datasets* $S$, $S'$ of size $n$, for any subset $T \subseteq \Theta$ it holds that $\Pr[\mathcal{T}(S) \in K] \le \Pr[\mathcal{T}(S') \in K] + \delta$.

It is easy to see that $(\epsilon, \delta)$-DP immediately implies $\delta'$-TV stability with:

$$\delta' = \exp(\epsilon) - 1 + \delta. \tag{2}$$

**From Classical to Distributional Generalization.** Similarly to the classical generalization, one way to achieve distributional generalization is through strong stability:

**Theorem 2.5.** *Suppose that the training algorithm is $\delta$-TV stable. Then, the algorithm satisfies $\delta$-DG.*

We refer to Appendix B for the proofs of this and all other formal statements in the rest of the paper.

As DP implies TV-stability, by Theorem 2.5 we have that DP also implies DG. We show that DP algorithms enjoy a significantly stronger stability guarantee than previously known, which means that the DG guarantee that one obtains from DP is also stronger.

**Proposition 2.6.** *An algorithm which is $(\epsilon, \delta)$-DP is also $\delta'$-TV stable with:*

$$\delta' = \frac{\exp(\epsilon) - 1 + 2\delta}{\exp(\epsilon) + 1}.$$

In Appendix A.2, we discuss the relationship of this result to other works in the literature on information-theoretic generalization. In particular, to Steinke and Zakynthinou [2020] whose results can also be used to relate DP and DG. Figure 2 shows that the known bounds quickly become vacuous unlike the bound in Proposition 2.6. In fact, we show that our bound is tight in Appendix B.

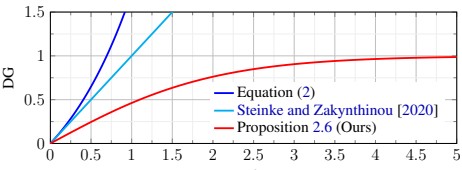

Figure 2: DG bound from $(\epsilon, 0)$-DP.

**Stronger Distributional Generalization Guarantees.** Although DG immediately implies generalization for all bounded properties, it is possible to obtain tighter bounds from TV stability. For example, directly applying $\delta$-DG to the *subgroup loss* property yields a bound that decays with the size of the subgroup: accuracy on very small subgroups is not guaranteed to generalize well. In Appendix C.1 we show that TV stability in fact implies "subgroup DG", which guarantees that the accuracy on even small subgroups generalizes well in expectation. As another example, in Appendix C.2 we show that TV stability also ensures the generalization of calibration properties of the learning algorithm.

## 3 Example Applications

To demonstrate that WYSIWYG is a useful property in algorithm design, in the remainder of this paper we use it to construct simple and high-performing algorithms for three example applications: mitigation of disparate impact of DP, ensuring group-distributional robustness, and mitigation of robust overfitting in adversarial training.

**Mitigating Disparate Impact of DP.** First, we consider applications in which learning presents privacy concerns, e.g., in the case that the training data contains sensitive information. Using training procedures that satisfy DP is a standard way to guarantee privacy in such settings. Training with DP, however, is known to incur *disparate impact* on the model accuracy: some subgroups of inputs can have worse test accuracy than others. For example, Bagdasaryan et al. [2019] show that using DP-SGD—a standard algorithm for satisfying DP [Abadi et al., 2016]—in place of regular SGD causes a significant accuracy drop on "darker skin" faces in models trained on the CelebA dataset of celebrity faces [Liu et al., 2015], but a less severe drop on "lighter skin" faces. Our goal is to mitigate such disparate impact. This issue—a quality-of-service harm [Madaio et al., 2020]—is but one of many possible harms due to ML systems. We do not aim to mitigate any other broad fairness-related issues, nor claim this is possible within our framework.

Formally, we assume the data distribution $\mathcal{D}$ is a mixture of $m$ groups indexed by set $\mathcal{G} = \{1, \ldots, m\}$, such that $\mathcal{D} = \sum_{i=1}^{m} q_i \mathcal{D}_i$. The vector $(q_i, \ldots, q_m) \in [0, 1]^m$ represents the group probabilities, with $\sum_{i=1}^{m} q_i = 1$. For given parameters $(\epsilon, \delta)$, we want to learn a model $\theta$ that simultaneously satisfies $(\epsilon, \delta)$-DP, has high overall accuracy, and incurs small *loss disparity*:

$$\max_{g, g' \in \mathcal{G}} \left| \mathop{\mathbb{E}}_{z \sim \mathcal{D}_g} [\ell(z; \theta)] - \mathop{\mathbb{E}}_{z \sim \mathcal{D}_{g'}} [\ell(z; \theta)] \right|. \tag{3}$$

**Group-Distributional Robustness.** Next, we consider a setting of *group-distributionally robust optimization* [e.g., Sagawa et al., 2019a, Hu et al., 2018]. If in the standard learning approach we want to train a model that minimizes *average* loss, in this setting, we want to minimize the *worst-case (highest) group loss*. This objective can be used to mitigate fairness concerns such as those discussed previously, as well as to avoid learning spurious correlations [Sagawa et al., 2019a].

Formally, we want to learn a model $\theta$ that minimizes the *worst-case group loss*:

$$\max_{g \in \mathcal{G}} \mathop{\mathbb{E}}_{z \sim \mathcal{D}_g} [\ell(z; \theta)]. \tag{4}$$

Unlike the previous application, in this setting, we do not require privacy of the training data. We use training with DP as a *tool* to ensure the generalization of the worst-case group loss.

**Mitigating Robust Overfitting.** Finally, we consider the setting of robustness to test-time adversarial examples through adversarial training [Goodfellow et al., 2014]. A common way to train robust models in this sense in image domains is to minimize *robust (adversarial) loss* [Madry et al., 2018]:

$$\mathop{\mathbb{E}}_{(x, y) \sim \mathcal{D}} \left[ \max_{\|x - x_{\mathsf{adv}}\|_p \leq \gamma} \ell((x_{\mathsf{adv}}, y); \theta) \right], \tag{5}$$

---
**Algorithm 1** DP-IS-SGD (DP Importance Sampling SGD)
---
**Input:** Dataset $S$, loss $\ell(z; \theta)$, initial parameters $\theta_0$, learning rate $\eta$, maximal gradient norm $C$, noise parameter $\sigma$, number of epochs $T$, sampling rate $\bar{p}$, group probabilities $(q_1, \ldots, q_m)$ .

 **for** $t = 1, \ldots, T$ **do**

  Sample batch $S_t \leftarrow \text{Sample}_{p(\cdot)}(S)$, with sampling probabilities $p(z) \triangleq \bar{p}/m \cdot q_{g(z)}$

  $\tilde{g}_t \leftarrow \frac{1}{|S_t|} \sum_{z \in S_t} \underbrace{1/\max\{1, \, C^{-1} \cdot \|\nabla_\theta \ell(z;\theta)\|_2\}}_{\text{Gradient clipping}} \cdot \nabla_\theta \ell(z; \theta) + \underbrace{\sigma C \cdot \mathcal{N}(0, I)}_{\text{Gradient noise}}$

  $\theta_t \leftarrow \theta_{t-1} + \eta \cdot \tilde{g}_t$
---

The highlighted parts indicate the differences with respect to DP-SGD. We obtain DP-SGD as a special case when we have a single group with $q = 1$ (implying $p(z) = \bar{p}$).

where $\gamma > 0$ is a small constant. Rice et al. [2020] observed that adversarially trained models exhibit "robust overfitting": higher generalization gap of robust loss than that of the regular loss. In this application, we similarly aim to use a relaxed version of training with DP as a tool to ensure generalization of robust loss, thus mitigate robust overfitting.

## 4 Algorithms which Distributionally Generalize

In this section, we construct algorithms for the applications in Section 3. Our approach follows the blueprint: First, we apply a principled algorithmic intervention that ensures desired behavior on *the training data* (e.g., importance sampling). Second, we modify the resulting algorithm to additionally ensure DG, which guarantees that the desired behavior generalizes to *test time*.

### 4.1 DP Training with Importance Sampling

Our first algorithm, DP-IS-SGD (Algorithm 1), is a version of DP-SGD [Abadi et al., 2016] which performs importance sampling. DP-IS-SGD is designed to mitigate disparate impact while retaining DP guarantees. The standard DP-SGD samples data batches using *uniform Poisson subsampling:* Each example in the training set is chosen into the batch according to the outcome of a Bernoulli trial with probability $\bar{p} \in [0, 1]$. To correct for unequal representation and the resulting disparate impact, we use *non-uniform Poisson subsampling:* Each example $z \in S$ has a possibly different probability $p(z)$ of being selected into the batch, where $p(z)$ does not depend on the dataset $S$ otherwise, and is bounded: $0 \le p(z) \le p^* \le 1$. We denote this subsampling procedure as $\text{Sample}_{p(\cdot)}(S)$.

We assume that we know to which group any $z = (x, y)$ belongs, denoted as $g(z)$, e.g., the group is one of the features in $x$. We choose $p(z)$ to satisfy two properties. First, to increase the sampling probability for examples in minority groups: $p(z) \propto 1/q_{g(z)}$. Second, to keep the average batch size equal to $\bar{p} \cdot n$ as in standard DP-SGD. In the rest of the paper, we assume that the group probabilities $(q_1, \ldots, q_m)$ are known, but it is possible to estimate them in a private way using standard methods [Nelson and Reuben, 2020]. We present DP-IS-SGD in Algorithm 1, along with its differences to the standard DP-SGD.

**DP Properties of DP-IS-SGD.** Uniform Poisson subsampling is well-known to amplify the privacy guarantees of an algorithm [Chaudhuri and Mishra, 2006, Li et al., 2012]. For example, Li et al. [2012] show that if an algorithm $\mathcal{T}(S)$ satisfies $(\epsilon, \delta)$-DP, then $\mathcal{T} \circ \text{Sample}_{\bar{p}}(S)$ provides approximately $(O(\bar{p}\epsilon), \bar{p}\delta)$-DP for small values of $\epsilon$. We show in Appendix B that non-uniform Poisson subsampling provides the same amplification guarantee with $\bar{p} = p^*$, where $p^*$ is the maximum value of $p(\cdot)$.

As this guarantee is independent of the internal workings of $\mathcal{T}(S)$, it is loose. For DP-SGD, one way of computing tight privacy guarantees of subsampling is using the notion of *Gaussian differential privacy* (GDP) [Dong et al., 2019]. GDP is parameterized by a single parameter $\mu$. If an algorithm $\mathcal{T}(S)$ satisfies $\mu$-GDP, one can efficiently compute a set of $(\epsilon, \delta)$-DP guarantees also satisfied by $\mathcal{T}(S)$ [Dong et al., 2019]. We show that we can use any GDP-based mechanism for computing the privacy guarantee of DP-SGD to obtain the privacy guarantees of DP-IS-SGD in a black-box manner:

**Proposition 4.1.** *Let us denote by $\mu(\bar{p}, \sigma, C, T)$ (see Algorithm 1) a function that returns a $\mu$-GDP guarantee of DP-SGD. Then, DP-IS-SGD satisfies a GDP guarantee $\mu(p^*, \sigma, C, T)$.*

## 4.2 Gaussian Gradient Noise

We showed that DP-IS-SGD enjoys theoretical guarantees for both DP and DG. DP models, however, often have lower test accuracy compared to standard training [Chaudhuri et al., 2011]. This can be an unnecessary disadvantage in settings where privacy is not required, such as in our robustness applications. Thus, we explore training algorithms which are inspired by our theory yet do not come with generic theoretical guarantees of DG.

Note that DP-SGD uses gradient *clipping* and *noise* (see Algorithm 1). Individually, these are used as *regularization methods* for improving stability and generalization [Hardt et al., 2016, Neelakantan et al., 2015]. Following this, we relax DP-IS-SGD to only use gradient noise. This sacrifices privacy guarantees in exchange for practical performance. Specifically, we apply gradient noise to three standard algorithms for achieving group-distributional robustness: importance sampling (IS-SGD), importance weighting (IW-SGD) [Gretton et al., 2009], and gDRO [Sagawa et al., 2019a]. This results in the following variations: IS-SGD-n, IW-SGD-n, gDRO-n, respectively. Similarly, we apply gradient noise to standard PGD adversarial training [Madry et al., 2018]. See Appendix D for details.

# 5 Experiments

We empirically study the distributional generalization in real-world applications. The code for the experiments is available at `https://github.com/yangarbiter/dp-dg`.

**Datasets.** We use the following datasets with group annotations: CelebA [Liu et al., 2015], UTK-Face [Zhang et al., 2017], iNaturalist2017 (iNat) [Horn et al., 2018], CivilComments [Borkan et al., 2019], MultiNLI [Williams et al., 2018, Sagawa et al., 2019a], and ADULT [Kohavi et al., 1996]. For every dataset, each example belongs to one group (e.g., CelebA) or multiple groups (e.g., Civil-Comments). For example, in the CelebA dataset, there are four groups: "blond male", "male with other hair color", "blond female", and "female with other hair color". Additionally, we use the CIFAR-10 [Krizhevsky et al., 2009] dataset for the adversarial-overfitting application. We present more details on the datasets, their groups, and used model architectures in Appendix E.

## 5.1 Enforcing DG in Practice

We empirically confirm that a training procedure with DP guarantees also has a bounded DG gap.

In practice, it is not possible to compute the exact DG gap. As a proxy in applications which concern subgroup performance in this section, and Sections 5.2 and 5.3, we use the difference between train-time and test-time worst-group accuracy. This (a) follows the empirical approach by Nakkiran and Bansal [2020] which proposes to estimate the gap in Equation (1) using a finite set of test functions, and (b) measures the aspect of distributional generalization that is relevant to our applications. We provide more details on this choice of the proxy measure in Appendix E.2.

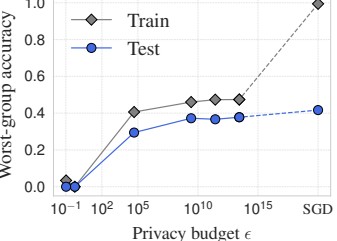

We train a model on CelebA using DP-SGD for varying levels of $\epsilon$. Figure 3 shows that the gap between training and testing worst-group accuracy increases as the level of privacy decreases, which is consistent with our theoretical bounds. In Appendix E.3 we also explore how regularization methods which do not necessarily formally imply DG, can empirically improve DG.

Figure 3: **Privacy induces DG.** Train/test worst-case group accuracies as a function of privacy parameter $\epsilon$ of DP-SGD on CelebA (x axis). Increasing privacy reduces the generalization gap.

## 5.2 Disparate Impact of Differentially Private Models

We evaluate DP-IS-SGD (Algorithm 1), and demonstrate that it can mitigate the disparate impact in realistic settings where both privacy and fairness are required.

Figure 4 shows the accuracy disparity, test accuracy, and worst-case group accuracy, computed as in Equation (3), as a function of the privacy budget $\epsilon$. The models are trained with DP-SGD and DP-IS-SGD. When comparing DP-SGD and DP-IS-SGD with the same or similar $\epsilon$, we observe that DP-IS-SGD achieves lower disparity on all datasets. However, this comes with a drop in average

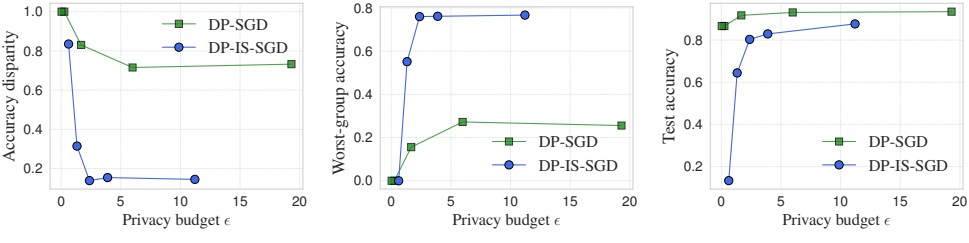

(a) Accuracy disparity (lower is better)   (b) Worst-group accuracy (higher is better)   (c) Test accuracy (higher is better)

Figure 4: **Importance Sampling Improves Disparate Impact of DP-SGD.** The accuracy disparity of the models trained with DP-SGD and DP-IS-SGD on CelebA. Adding importance sampling (IS) improves disparate impact at most privacy budgets in this setting. We set $\delta = 1/2n$, where $n$ is the number of training examples. We use GDP accountant to compute the privacy budget $\varepsilon$.

accuracy. On CelebA, with $\epsilon \in [2, 12]$, DP-IS-SGD has around 8 p.p. lower test accuracy than DP-SGD. At the same time, the disparity drop ranges from 40 p.p. to 60 p.p., which is significantly higher than the accuracy drop. We observe similar results on UTKFace. On iNat, however, although DP-IS-SGD decreases disparity, the overall test accuracy suffers a significant hit. This is likely because the minority subgroup is very small, which results in high maximum sampling probability $p^*$, thus deteriorating the privacy guarantee. Details for UTKFace and iNat are in Appendix E.4.

In summary, we find that DP-IS-SGD can achieve lower disparity at the same privacy budget compared to standard DP-SGD, with mild impact on test accuracy.

**Comparison to DP-SGD-F [Xu et al., 2021].** DP-SGD-F is a variant of DP-SGD which dynamically adapts gradient-clipping bounds for different groups to reduce the disparate impact. We did not manage to achieve good overall performance of DP-SGD-F on the datasets above. In Appendix E.4, we compare it to DP-IS-SGD on the ADULT dataset (used by Xu et al. [2021]), finding that DP-IS-SGD obtains lower disparity for the same privacy level, yet also lower overall accuracy.

## 5.3   Group-Distributionally Robust Optimization

We investigate whether our proposed versions of standard algorithms with Gaussian gradient noise (Section 4.2) can improve group-distributional robustness. To do so, we evaluate empirical DG using worst-group accuracy as a proxy for DG gap as in Section 5.1, following the evaluation criteria in prior work [Sagawa et al., 2019a, Idrissi et al., 2022]. State-of-the-art (SOTA) methods apply $\ell_2$ regularization and early-stopping to achieve the best performance. We compare three baselines with $\ell_2$ regularization, IS-SGD-$\ell_2$, IW-SGD-$\ell_2$, and gDRO-$\ell_2$ to our noisy-gradient variations as well as DP-IS-SGD. We use the validation set to select the best-performing regularization parameter and epoch (for early stopping) for each method. See Appendix E.5 for details on the experimental setup.

Table 1 shows the worst-group accuracy of each algorithm on five datasets. When comparing IS-SGD, IW-SGD, and gDRO with their noisy counterparts, we observe that the noisy versions in general have similar or slightly better performance compared to non-noisy counterparts. For instance, IS-SGD-n improves the SOTA results on CivilComments dataset. This showcases that in terms of learning distributionally robust models, *noisy gradient can be a more effective regularizer than the currently standard $\ell_2$ regularizer*. We also find that DP-IS-SGD improves on baseline methods or even achieves SOTA-competetitive performance on several datasets. For instance, on CelebA and MNLI, DP-IS-SGD achieves better performance than IS-SGD-$\ell_2$. This is surprising, as DP tends to deteriorate performance. This suggests that distributional robustness and privacy are not incompatible goals. Moreover, DP can be a useful tool even when privacy is not required.

## 5.4   Mitigating Robust Overfitting

As in the previous section, we expect that a modification of a standard projected gradient descent (PGD) method for adversarial training [Madry et al., 2018] with added Gaussian gradient noise (Section 4.2) improves the generalization behavior of adversarial training.

---

[1]IW-SGD numbers are different from Fig. 1, as in the figure we do not apply regularization.

Table 1: **Our noisy-gradient algorithms produce competitive results compared to counterparts with $\ell_2$ regularization.** The table shows the worst-group accuracy of each algorithm. Baselines are in the top rows; our algorithms are in the bottom. For gDRO-$\ell_2$-SOTA, we show avg. $\pm$ std. over five runs from Idrissi et al. [2022]. For CelebA, we show avg. $\pm$ std. over three random splits.

| | CelebA | UTKFace | iNat. | Civil. | MNLI |
|---|---|---|---|---|---|
| SGD-$\ell_2$ | $73.0 \pm 2.2$ | 86.3 | 41.8 | 57.4 | 67.9 |
| IS-SGD-$\ell_2$ | $82.4 \pm 0.5$ | 85.8 | 70.6 | 64.3 | 70.4 |
| IW-SGD-$\ell_2$[1] | $\mathbf{89.0} \pm 0.9$ | 86.5 | 67.6 | 65.7 | 68.1 |
| gDRO-$\ell_2$ | $84.5 \pm 0.8$ | 85.2 | 67.3 | 67.3 | 75.9 |
| gDRO-$\ell_2$-SOTA | $86.9 \pm 0.5$ | — | — | $69.9 \pm 0.5$ | $\mathbf{78.0} \pm 0.3$ |
| DP-IS-SGD | $86.0 \pm 0.8$ | 82.5 | 51.4 | 70.4 | 72.3 |
| IS-SGD-n | $84.9 \pm 1.0$ | 85.5 | $\mathbf{71.0}$ | $\mathbf{71.9}$ | 70.8 |
| IW-SGD-n | $\mathbf{88.5} \pm 0.4$ | $\mathbf{88.5}$ | 70.9 | 69.9 | 69.7 |
| gDRO-n | $83.3 \pm 0.5$ | 87.5 | 56.4 | 71.3 | $\mathbf{78.0}$ |

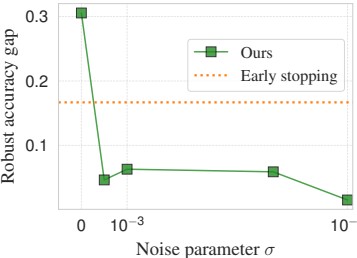

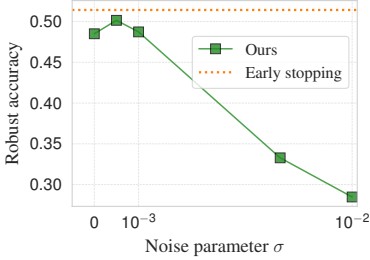

(a) Gen. gap of robust accuracy (lower is better)      (b) Robust accuracy (higher is better)

Figure 5: **Noisy gradient reduces overfitting in adversarial training.** We show the generalization gap of robust accuracy (left), and test-time robust accuracy (right) of adversarially trained models with different levels of noise magnitude. The dash orange lines represent the performance of adversarial training with early stopping. The model trained without noise exhibits "robust overfitting" of about 30 p.p. Gradient noise reduces the generalization gap by more than $3\times$ for all values of the noise parameter at a cost of decreased robust accuracy as the noise gets larger.

To verify this, we adversarially train models on the CIFAR-10 dataset with varying levels of the noise magnitude. We provide more details on the setup in Appendix E.6. Figure 5 shows that in standard adversarial training without noise the gap between robust training accuracy and robust test accuracy is large at approximately 30 p.p., which is consistent with the prior observations of Rice et al. [2020]. By injecting noise into the gradient, our proposed approach decreases the generalization gap of robust accuracy by more than $3\times$ to less than 10 p.p. Surprisingly, in our experiments, training with gradient noise achieves both a small adversarial accuracy gap *and* better adversarial test accuracy compared to standard adversarial training, when using a small noise magnitude ($\sigma = 5 \times 10^{-4}$). In terms of resulting robust accuracy, the method's performance is comparable to early stopping, identified as the most effective way to prevent robust overfitting by Rice et al. [2020]. These experimental results demonstrate how WYSIWYG can be a useful design principle in practice.

# 6  Related Work

**DP and Strong Generalization.** DP is known to imply a stronger than standard notion of generalization, called *robust generalization*[2] [Cummings et al., 2016, Bassily et al., 2016]. Robust generalization can be thought as a high-probability counterpart of DG: generalization holds with high probability over the train dataset, not only on average over datasets. We focus on our notion of DG for both conceptual and theoretical simplicity. A more comprehensive discussion of relations to robust generalization is in Appendix A.1. It can also be useful to consider intermediary definitions, varying in strength from DG to robust generalization. In Appendix C, we introduce such an notion ("strong DG") and show its connections to DP. Other than robust generalization, our connections in

---

[2]Unrelated to "robust overfitting" in adversarial training.

Section 2 can also be derived from weaker generalization bounds that rely on information-theoretic measures [Steinke and Zakynthinou, 2020]. We detail this in Appendix A.2.

**Disparate Impact of DP.** Bagdasaryan et al. [2019], Pujol et al. [2020] have shown that ensuring DP in algorithmic systems can cause error disparity across population groups. Xu et al. [2021] proposed a variant of DP-SGD for reducing disparate impact. We compare our method to DP-SGD-F in Appendix E.4. In another line of related work, Sanyal et al. [2022], Cummings et al. [2019] show fundamental trade-offs between model's loss and DP training. As our theoretical results concern generalization, not loss per se, our results do not contradict these theoretical trade-offs. We discuss the relationship in detail in Appendix A.3.

**Group-Distributional Robustness.** Group-distributional robustness aims to improve the worst-case group performance. Existing approaches include using worst-case group loss [Mohri et al., 2019, Sagawa et al., 2019a, Zhang et al., 2020], balancing majority and minority groups by reweighting or subsampling [Byrd and Lipton, 2019, Sagawa et al., 2019b, Idrissi et al., 2022], leveraging generative models [Goel et al., 2020], and applying various regularization techniques [Sagawa et al., 2019a, Cao et al., 2019]. Although some work [Sagawa et al., 2019a, Cao et al., 2019] discusses the importance of regularization in distributional robustness, they have not explored potential reasons for this (e.g. via the connection to generalization). Another line of work studies how to improve group performance without group annotations [Duchi et al., 2021, Liu et al., 2021, Creager et al., 2021], which is a different setting from ours as we assume the group annotations are known.

**Robust Overfitting.** Rice et al. [2020], Yu et al. [2022] have shown that adversarially trained models tend to overfit in terms of robust loss. Rice et al. [2020] proposed to use regularization to mitigate overfitting, but the noisy gradient has not been explored for this. We showed that the WYSIWYG framework can serve as an alternative direction for mitigating and explaining this issue.

# 7  Conclusions and Future Work

We argue that a "What You See is What You Get" property, which we formalize through the notion of distributional generalization (DG), can be desirable for learning algorithms, as it enables principled algorithm design in settings including deep learning. We show that this property is possible to achieve with DP training. This enables us to leverage advances in DP to enforce DG in many applications.

We propose enforcing DG as a general design principle, and we use it to construct simple yet effective algorithms in three settings. In certain fairness settings, we largely mitigate the disparate impact of differential privacy by using importance sampling and enforcing DG in our new algorithm DP-IS-SGD. In our analysis, however, the privacy and DG guarantees of DP-IS-SGD deteriorate in the presence of very small groups. Future work could explore individual-level accounting [Feldman and Zrnic, 2021] for a tighter analysis. In certain worst-case generalization settings, inspired by DP-SGD, we propose using a noisy-gradient regularizer. Compared to SOTA algorithms in DRO, noisy gradient achieves competitive results across many standard benchmarks. In certain adversarial-robustness settings, our proposed noisy-gradient regularizer significantly reduces robust overfitting. An interesting direction for future work would be to explore its effectiveness in large-scale settings, e.g., ImageNet [Croce and Hein, 2022]. We hope future work can explore extending this design principle to ensure generalization of other properties, such as calibration and counterfactual fairness.

**Acknowledgements**

We thank Kamalika Chaudhuri, Benjamin L. Edelman, Saurabh Garg, Gautam Kamath, Maksym Andriushchenko, and Cassidy Laidlaw for useful feedback on an early draft. BK acknowledges support from the Swiss National Science Foundation with grant 200021-188824. Yao-Yuan Yang thanks NSF under CNS 1804829 and ARO MURI W911NF2110317 for the research support. Yaodong Yu acknowledges support from the joint Simons Foundation-NSF DMS grant #2031899. PN is grateful for support of the NSF and the Simons Foundation for the Collaboration on the Theoretical Foundations of Deep Learning[3] through awards DMS-2031883 and #814639.

---

[3] https://deepfoundations.ai

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
