# OpenReview forum: "What You See is What You Get: Principled Deep Learning via Distributional Generalization"
_NeurIPS.cc/2022/Conference — NeurIPS 2022 Accept_

### Official Review · Reviewer_XhyC · 2022-07-05

**Rating:** 8
**Confidence:** 5
**Soundness:** 3 good
**Presentation:** 4 excellent
**Contribution:** 4 excellent

**Summary:**

This work proposes to measure a model’s generalization under distribution shift with the previously proposed distributional generalization (DG), which is the maximum generalization error of the model under any loss function within [0,1]. They then prove that differential privacy (DP) leads to TV stability, which further leads to DG. Motivated by this result, the authors propose the DP-IS-SGD algorithm, which is essentially an importance weighted version of DP-SGD, so it requires the group labels at train time. However, DP usually leads to models with lower test performances, so the authors also experiment with adding noisy gradients to robust training methods like importance weighting and Group DRO.

**Questions:**

1. Definitions 2.1 and 2.2 are confusing. In Eqn. (1), use a different notation: Change z ~ S to zhat ~ S to make it clear that these are the training samples, and this is the gap between training error and test error. Also in Definition 2.2 pi(Z) should be pi(Z; theta(S)).
2. Since DG is very hard to achieve, I suppose that the algorithms the authors propose cannot achieve good DG either. So is DG just an indicator of robust generalization and cannot be really maximized? Can we somehow find some “surrogate loss” for DG or DP that can be optimized more easily?


**Limitations:**

The limitations are addressed in Section 3.

**Strengths And Weaknesses:**

Overall I find that there is a strong case to accept this work. The issue that popular methods such as importance weighting cannot generalize well is a well-known problem in this field, and this work investigates this problem through a theoretical lens and proposes some practical algorithms. The strengths of this work are:

1. The theory part is very well written and easy to understand. It nicely builds a bridge between DP and distributional robustness.
2. The algorithms make a lot of sense, and they show that noisy gradients can be a better regularizer than standard L2 regularization.

The biggest issue is that the DG condition seems too strong and very hard to achieve in practice. If I understand correctly, delta-DG implies (delta, pi)-DG for all pi because we can set phi(Z;theta) to be Pr(pi(Z;theta)) (correct me if I am wrong). Thus, delta-DG is a very strong condition, given that (delta, pi)-DG covers fairness, adversarial robustness and so on. And since DP and TV stability can imply DG, it implies that DP and TV stability are also very strong and are generally hard to achieve in practice. Nevertheless, there is no doubt that this work points to a very interesting and promising direction for theory.

---

> ### Author Response · Authors · 2022-08-02
> **Response to Reviewer XhyC**
>
> > The DG condition seems too strong…
>
> It is a very strong condition, and this is why we are very excited about this work. We show that this strong condition can be provably guaranteed using standard and widely available methods of differentially-private training.
>
> > …and very hard to achieve in practice.
>
> Training with differential privacy is a rich area of research with multiple algorithms in the literature, that are also accessible via easy-to-use open-source frameworks such as opacus or diffprivlib. In this sense, we show that DG is in fact pretty “easy” to achieve in practice. It is “hard” in a different sense: The strong generalization guarantees are not free as training with differential privacy comes at a cost of reduced average performance, especially for deep learning models.
>
> > Since DG is very hard to achieve, I suppose that the algorithms the authors propose cannot achieve good DG either. So is DG just an indicator of robust generalization and cannot be really maximized?
>
> We show that training with differential privacy provably ensures a certain level of DG. We do not, however, explicitly maximize DG. It should be theoretically possible to find randomization mechanisms (instead of the Gaussian noise used by DP-SGD) that would maximize DG in a more direct way. We will discuss this as a direction for future work.
>
> We thank the reviewer for the suggestions on how to make our notation clearer.

---

### Official Review · Reviewer_cawT · 2022-07-06

**Rating:** 5
**Confidence:** 3
**Soundness:** 3 good
**Presentation:** 3 good
**Contribution:** 2 fair

**Summary:**

The paper considers the “what you see is what you get” principle which means that the performance on the train dataset is similar to the performance on the test set. This principle is obtained from distributional generalisation which ensures that values of bounded functions are close on train and test sets. The paper then shows that by imposing differential privacy, distributional generalisation is achieved and uses this connection to define algorithms that maintain similar performance on train and test sets.

**Questions:**

See above in the cons. The main comment is around why we need the concept of DG.

**Limitations:**

There is no limitations section and this would be interesting to include.

**Strengths And Weaknesses:**

Pros:
- The notion that ensuring DG implies that behavior generalises to the test data is interesting.
- The concept that regularization methods improve DG is interesting (at least numerically).
- The paper is well-written and I enjoyed reading it.
- The numerical results are interesting.

Cons:
- As the authors mention, the privacy -> generalisation implication is well-known. The DP-> DG implication therefore seems to be a (smaller) extension of this known result.
- Since DP implies DG, why do we need this intermediate notion of DG? This is unclear to me. Algorithm 1 is essentially DP with importance sampling; it seems we do not make any modifications to achieve DG or use the properties of DG to define this algorithm (please correct me if I am wrong). Does DG give us any more information on what kind of noise to add and how to better trade off between noise/accuracy? This is thus my main comment: it is unclear what DG gives us in terms of algorithm design since the algorithms use DP; the case for DG is not entirely convincing. I can see that it allows us to derive the “what you see is what you get” guarantee, but this seems to also be derivable from DP itself.
- Section 4.2 could be explained in more detail - these algorithms thus do not do gradient clipping and only add noise, so that formally both DP and DG are not satisfied? This again makes me question why DG is relevant to have as a concept.
- As mentioned in 5.1 it is not possible to compute the exact DG gap so also numerically the notion of DG seems to be tricky to work with.

---

> ### Author Response · Authors · 2022-08-02
> **Response to Reviewer cawT**
>
> > The DP-> DG implication is a (smaller) extension of known results.
>
> Indeed, we discuss the relationship to the similar known results in Section 6 and Appendix A. Prior to our work, however, these results have not been applied in the way we propose in our WYSIWYG framework: For ensuring generalization of train-time interventions that are known to not generalize well for modern overparameterized models.
>
> > Since DP implies DG, why do we need this intermediate notion of DG?
>
> DG is one possible way to characterize the distributional generalization properties of training algorithms which has been considered in prior literature (see Section 2.1). As we explain in Section 6, paragraph “DP and Strong Forms of Generalization”, and Appendix A, we choose this notion for its simplicity and intuitive operational meaning. The conceptual property of WYSIWYG can be described using other formalisms, and we do so in Appendix C.
>
> Differential privacy, however, is a much stronger property than DG. As we show in Section 2, DG is implied by TV stability which is even weaker than DP (for example, it is often not considered a good enough notion of privacy as it allows “catastrophic” failures of privacy, see, e.g., [Barber and Duchi, 2014](https://arxiv.org/abs/1412.4451)). Training with differential privacy is but one possible and provable way to achieve DG.
>
> That said, we focus on standard training with differential privacy for most of our experiments, because it is a rich area of research with multiple algorithms in the literature, that are also accessible via easy-to-use open-source frameworks such as opacus or diffprivlib. We show that any of this vast body of algorithms immediately implies strong guarantees of DG, and propose a framework on how to practically make use of this theoretical connection.
>
> > It is unclear what DG gives us in terms of algorithm design since the algorithms use DP.
>
> As we describe in Section 4.2, our usage of Gaussian gradient noise regularizer exactly aims to find the intermediate point between DP which might be too strong for some applications, and DG. It is possible to design randomization mechanisms that would explicitly optimize DG, which we think is a promising yet challenging direction for future work (see also the [response to Reviewer XhyC](https://openreview.net/forum?id=g05fHAvNeXx&noteId=4gKM7hHD5xM)).
>
> > The notion of DG seems to be tricky to work with.
>
> Indeed, directly measuring the DG gap is challenging as it is defined for all bounded test functions. As we detail in Appendix E.2, there are different approaches to performing this measurement. Specifically, Nakkiran and Bansal, 2020 propose to evaluate the gap at a finite set of test functions. Another approach is running membership inference attacks (MIAs), whose average success rate is shown to be equivalent to the theoretical DG gap in the worst case (Kulynych et al. 2022). We tried MIAs, but for our tasks the standard attacks were not effective in the following sense: the gap estimated through MIAs was lower or equal than the gaps estimated using simpler test functions. Therefore, as we mention in Section 5.1, we take the approach by Nakkiran and Bansal of measuring the generalization gap of application-specific properties that are relevant to our tasks.

---

### Official Review · Reviewer_QRKD · 2022-07-12

**Rating:** 5
**Confidence:** 5
**Soundness:** 3 good
**Presentation:** 3 good
**Contribution:** 2 fair

**Summary:**

The paper proposes the WYSIWYG framework which simply, formally, captures the idea that all behaviors should be nearly identical on the train and test set. The paper further goes on to show that differentially private algorithms exhibit this property and then provides some experiment to show it in practice.

**Questions:**

* The authors mention that DP training can improve fairness issues/accuracy disparity as observed in Bagdasaryan et al [1] and Pujol et al [2]. However, I imagine the results do not extend to the case of Sanyal et. al. [3] where the number of minority subpopulations scales with the sample size and also Cummings et. al [4] which has a somewhat related setting too. I don't say this as a criticism but rather for a better understanding of the problem. Could the authors please cite these two papers and discuss it?


* In light of the previous point, I find the theoretical part of this paper to not be the overwhelmingly major component here. Further, the DG framework is a very strong form of generalisation as is mentioned by the authors and the experiments do not reflect that as I mentioned in my first point on weakness. The paper mentions a)  Counterfactual fairness, b)Robustness to corruptions, and  Adversarial robustness as further use cases of these. I would like to see experiments on some if not all of these.

* Line 172 and Proposition 2.6 - The paper states that "DP algorithms enjoy a significantly stronger stability guarantee than the previously known one". Could the authors clarify and cite which previously known bound they are improving ? If it is the rather trivial $\mathrm{exp}(\epsilon)-1+\delta$ bound that is being improved, I would urge that this **not** be stated as an improvement over **known bounds** but rather another result; otherwise it sounds like an existing non-trivial result is being improved upon.

* The work by Feldman et. al [3] also introduces instance dependent privacy accounting which could possibly used for an algorithm like DP-IS-SGD. I would ask the authors to discuss the relation to this.


[1] Bagdasaryan, Eugene, Omid Poursaeed, and Vitaly Shmatikov. "Differential privacy has disparate impact on model accuracy." Advances in neural information processing systems 32 (2019).
[2] Pujol, David, et al. "Fair decision making using privacy-protected data." Proceedings of the 2020 Conference on Fairness, Accountability, and Transparency. 2020.
[3] Sanyal, Amartya, Yaxi Hu, and Fanny Yang. "How unfair is private learning?." Proceedings of The 38th Conference on Uncertainty in Artificial Intelligence, 2022, https://openreview.net/forum?id=H2V43wIj5g9.
[4] Cummings, Rachel, et al. "On the compatibility of privacy and fairness." Adjunct Publication of the 27th Conference on User Modeling, Adaptation and Personalization. 2019.
[5] Feldman, Vitaly, and Tijana Zrnic. "Individual privacy accounting via a renyi filter. arXiv preprint 247." arXiv preprint arXiv:2008.11193 248 (2020).

**Limitations:**

I have already discussed above that the paper does not address existing literature well.

**Strengths And Weaknesses:**

Strength

* The paper highlights an interesting framework to capture the notion that**all** properties should behave uniformly on the train and the test set. The paper captures this idea and expresses it very well.
* The experiments with DP-IS-SGD and G-DRO provides a beautiful proof-of-concept. However, as I mention below, I think this is merely a small proof of concept and this nice but extremely limited experiment is not sufficient, in my opinion.

Weakness

I have listed two main weaknesses here - a) the experiment is simply too limited to claim anything about the generality of the framework and the theory ignores a huge body of work that proves information-theoretic generalisation bounds that applies to DP algorithms and, in fact, also shows distributional generalisation. I have listed further questions in the Questions section below.

* To show applications of the framework, the claims to experiment on
  i. Group Distributionally Robust Optimisation and
  ii. mitigate the disparate impact of DP.
  However, in reality the first is a direct derivate of the second. More precisely, in the second case the authors propose an algorithm, derived from DP-SGD and that is guaranteed to be private with gradient clipping and adding noise *smartly*. In the the GDRO, the clipping part is removed. I find it quite a stretch to claim that the paper shows two applications of the framework.


* There is a large body of work on information theoretic generalisation bounds that is largely ignored by the authors. For example generalisation bounds based on bounded conditional mutual information (Steinke et. al [a]) is not discussed. In fact, the authors there also discuss their relation to DP and possibly their paper yields stronger results than this. On a more high level, there is now a huge body of work on generalisation guarantees of DP algorithms that are usually loss function agnostic and hence provides strong DG type guarantees. I would like to see a discussion of this.


[a] Steinke, Thomas, and Lydia Zakynthinou. "Reasoning about generalization via conditional mutual information."

---

> ### Author Response · Authors · 2022-08-02
> **Response to Reviewer QRKD**
>
> > Could the authors please cite Cummings et al. 2019 and Sanyal et al. 2022 and discuss it?
>
> Absolutely. We thank the reviewer for pointing us to the concurrent Sanyal et al. paper.
> We agree there should be a discussion of these theoretical results in our paper. We would like to point out that there is no tension between our theoretical results and theirs. Both these papers are about the relationship between privacy and disparate performance (accuracy or false-positive/false-negative rates), whereas we discuss the relationship between privacy and generalization _gap_. Even if a DP model has to incur at least a certain error on small subgroups on average (see Sanyal et al. 2022, Lemma 1), this error is guaranteed to be similar at train time and test time (from our theoretical discussion). There is no tension in the experimental part either. If we visualize the lower bound on subgroup error in Lemma 1 from Sanyal et al. 2022, we see that this inherent lower bound vanishes for subgroups of size greater than 100 even for small epsilons (e.g., 0.1). The subgroups and values of epsilon in our experiments are all larger than this, thus in our regime we can achieve meaningful subgroup performance with the DP-IS-SGD algorithm despite the fundamental trade-off.
>
> > The work by Feldman et. al [3] also introduces instance dependent privacy accounting which could possibly used for an algorithm like DP-IS-SGD. I would ask the authors to discuss the relation to this.
>
> We thank the reviewer for this pointer. Indeed, our accounting of DP-IS-SGD is worst-case, as our overall privacy guarantee for DP-IS-SGD depends on the worst-group privacy leakage (Lemma C.7 in our appendix). Individual-level or group-level accounting could make the analysis tighter. We will discuss this in the paper.
>
> > There is a body of literature on generalization based on information theory that the paper ignores, which could result in stronger bounds.
>
> We have discussed the relationship with the strongest possible generalization guarantees that one can obtain from DP – robust generalization – in Section 6, and, in more detail, in Appendix A. In the revision, we will also discuss the relationship to Steinke and Zakynthinou 2020. We can indeed use their results to bound DG from DP (from a combination of either Theorem 4.7/4.10 and Corollary 5.2 in their paper), but such conversions are looser, as our bound is tight.
>
> > Experiments are not enough to establish the generality of the framework.
>
> We assume that the reviewer means generality with respect to different properties within the WYSIWYG framework. In this paper, we focus on one — socially relevant and important — property of subgroup loss, for which we provide empirical support. We leave the empirical investigations of other hypothetical properties for future work, as we mention in Section 7. We will make this distinction clearer.
>
> We would like to note that our experimental support is fairly extensive: we experiment on multiple domains (images, language), and our results are beating SOTA in well-established benchmarks (see Table 1). For our privacy experiments, we sweep across different scales of privacy budget, to demonstrate that our methods perform well at different levels of privacy. We thus believe that our experiments provide broad support for our claims.
>
> > The two applications are the same application.
>
> Although we agree that the property in terms of the WYSIWYG framework is the same for both applications (subgroup loss), the two problems are different. They are not defined by us, but rather posed by the existing literature, and this is why we consider them (see, e.g., Bagdasaryan and Shmatikov, 2019; Sagawa et al, 2019). As described in Section 4, they differ in privacy requirements and relevant tasks, thus require different approaches.

---

> > ### Comment · Reviewer_QRKD · 2022-08-05
> > **Thanks for the response**
> >
> > I thank the authors for the detailed responses.
> >
> > * I still stand by my point that the concept of WYSIWYG is a beautiful concept but it is simply a **very strong theoretical notion** similar to other information-theoretic generalisation bounds. The only new theoretical concept here is that DP implies WYSIWYG, which in itself is also known from for example Steinke et. al. but the authors here provide a nice proof-of-concept experiment.
> >
> > * Given the point above, I would be really happy to advocate for this paper if only the authors showed the generality of this method in some more areas. The paper even mentions some areas it could be applied to but don't show any algorithms or experimental results.
> >
> > * The point about obtaining SOTA results, I must say, is really grasping at straws. In CELEBA, the DP method is worse and in MNLI the method is the same. In others, one of the DP methods perform better by 1 or 2% and other DP methods worse. I am happy to say "competitive" but not saying that they are beating SOTA.
> >
> > Having said that, I appreciate the discussion of the authors on the other points in the rebuttal and would like to see them written and updated in the paper.  I agree with the authors response that the results here don't contradict but rather nicely support the results in Sanyal et. al. 2022 and also Cummings et. al I think. I think a discussion of Feldman et. al. should also be included, illustrating what the advantages and disadvantages would be of doing instance dependent privacy accounting especially how it interacts with WYSIWYG framework. I would also like to understand why the results from Steinke et. al. would be looser. If all these discussions are included in the paper, I would be happy with the discussions in the paper and my main qualm would be about overclaiming the validation of the generality of the proposed notion.

---

> > > ### Author Response · Authors · 2022-08-09
> > > **Additional experiments and discussions in the revised version**
> > >
> > > ## Summary
> > >
> > > We thank the reviewer for the concrete pointers. Following the reviewer's suggestions, we uploaded a revised revision (main pdf, _not_ the supplementary material), which includes:
> > >
> > > - *Appendix A.2:* Discussion of the related work on information-theoretic generalization bounds, specifically the results of Steinke and Zakynthinou, 2020. As we detail and visualize (Fig. 5) in the revised text, the bound on DG from DP that we can derive from their results is much looser than our bound. We can, however, use our theoretical result in Proposition 2.6 to sharpen their bounds from DP to CMI in the $\epsilon \geq 1$ regime (CMI is the main object of study of Steinke and Zakynthinou, 2020). We do this in Corollary A.1. In this section, we also point out that individual-level notions of stability such as the one used by Feldman and Zrnic, 2020, could be used to tighten bounds on DG similarly to individual-level mutual information notions.
> > >
> > > - *Appendix A.3:* Discussion of the related work on theoretical trade-offs between privacy and fairness (see our [previous response](https://openreview.net/forum?id=g05fHAvNeXx&noteId=odh1C1S6rACs)).
> > >
> > > - *Appendix E.6:* New experimental results showing the generality of the framework, in which we use the noisy-gradient regularizer to reduce _overfitting in adversarial training_: the generalization gap of adversarial accuracy. We detail this experiment below.
> > >
> > > ## Other comments
> > >
> > > > I am happy to say "competitive" but not saying that they are beating SOTA.
> > >
> > > We apologize for claiming to “beat” in the previous response. “Competitive” is correct and is the formulation that we are using in the revised version.
> > >
> > > ## Details about the new experiments in Appendix E.6
> > >
> > > > I would be really happy to advocate for this paper if only the authors showed the generality of this method in some more areas. The paper even mentions some areas it could be applied to but don't show any algorithms or experimental results [in these other areas]
> > >
> > > Thank you for your suggestion on the experiments. We have conducted new experiments using a different property from subpopulation loss: adversarial loss (last item in our Section 2.1’s list of possible properties). Specifically, inspired by our theory, we applied SGD with noisy gradients to standard PGD adversarial training [Madry et al., 2018]. This is the same principle as we used in Section 4.2 (“Group-DRO with Noisy Gradients”). We used the CIFAR-10 dataset and ResNet-18 as the network architecture. We trained the model to be robust against $L_\infty$ perturbations with the $8/255$ bound, which is a standard setup for adversarial training on this dataset. We varied sigma (noise parameter) from 0.0 (regular adversarial training; no noise) to 0.01.
> > >
> > > We summarize the results in Figure 9 in the Appendix E.6. The left plot displays the robust accuracy gap ($|\textrm{robust train accuracy} - \textrm{robust test accuracy}|$) and the right plot displays the robust test accuracy. As shown in Figure 9 (a), without gradient noise, in standard adversarial training the gap between robust training accuracy and robust test accuracy is large at approximately 30 p.p. A similar phenomenon (the robust training accuracy and robust test accuracy is large) has been observed by Rice et al., 2020. By injecting noise into the gradient, our proposed approach decreases the robust accuracy gap by more than 3$\times$ to less than 10 p.p. More surprisingly, our approach can achieve both a low robust accuracy *gap* and better robust test accuracy compared to standard adversarial training, when using a low level of noise $\sigma=0.0005$. These experimental results further demonstrate how DG can be a useful design principle in practice.
> > >
> > > Thank you again for your suggestion. Because the revised version is limited to the same 9 pages as the submission version, the results have to be in the appendix. In the final version, we will add these results to the main text.
> > >
> > > [Rice et al., 2020] Leslie Rice, Eric Wong, J. Zico Kolter. Overfitting in adversarially robust deep learning. ICML2020.

---

### Official Review · Reviewer_4obB · 2022-07-12

**Rating:** 6
**Confidence:** 4
**Soundness:** 2 fair
**Presentation:** 3 good
**Contribution:** 3 good

**Summary:**

The paper applies the concept of distributional generalization (DG)  to mitigate unwanted behavior at test time. They consider two main applications: (a) mitigating disparate impact of differential privacy and (b) group-distributional robustness. The authors show that (the previously established notion of) DG leads to a “what you see is what you get” phenomena where effects (that can be characterized by a loss function, e.g. performance on a group, etc) observed in training data will translate to test data. Thus, the goal of the paper is to develop algorithms that satisfy DG and at the same time have good behavior on the underlying objective (e.g. good performance on all subgroups). In particular, there is a direct connection between privacy and generalization, as DP algorithms satisfy DG. The authors build on this insight to develop algorithms like DP-IS-SGD (for task-(a)) and a DP-SGD based noisy gradient regularizer (for task-(b)). As a result, the authors show improvements in objects (a) and (b) over DP-SGD, while retaining test loss performance.


**Questions:**

Is DP necessary to get DG? Are there concrete examples where DG provable holds but the algorithm is not private ?

**Limitations:**

The paper has no potential negative societal impact. In fact, the goal of the paper is to address various societal issues like (fairness, robustness, etc) observed in practice.

**Strengths And Weaknesses:**

Strength:
I think the direction explored in the paper is quite interesting, and having algorithmic principles which would imply that ensuring certain properties like robustness, fairness, caliberation, etc on train data is sufficient to ensure that these properties also transfer to the test data, is quite valuable. The paper provide experiments that justify their algorithms, and the underlyinig principle of “what you see is what you get”.

Weaknesses:
I think the paper is not very novel. The idea of distributional generalization and its connections to differential privacy was known before. While proposition 2.6 is new, I think the rest of the parts like DP-IS-SGD and DP-SGD with noisy gradient regularizer are straightforward extensions of known algorithmic ideas. It seems that the authors are just rebranding distributional generalization and differential privacy as “what you see is what you get” principle.

In general, it is not clear whether going though the route of designing algorithms with DG property is the best way to get desired behavior (fairness, robustness, etc) in practice. What are  the SOTA approaches for tasks-(a) and (b). It seems the experiments in Figure 3 or 4 does not compare to (or mention anything about) known SOTA approaches for mitigating these issues.

---

> ### Author Response · Authors · 2022-08-02
> **Response to Reviewer 4obB**
>
> > The idea of distributional generalization and its connections to differential privacy was known before.
>
> Indeed, we discuss the relationship to the similar known results in Section 6 and Appendix A. Prior to our work, however, these results have not been applied in the way we propose in our WYSIWYG framework: For ensuring generalization of train-time interventions, in particular those that are known to not generalize well for modern overparameterized models.
>
> > The experiments do not compare to (or mention anything about) known SOTA approaches for mitigating these issues.
>
> For task (b), this is not true. We compare against the state-of-the-art and other baselines, see Table 1.
>
> For task (a), as detailed in the Section 5.2 introduction, the question we were aiming to answer is whether our theoretic framework (algorithmic intervention to ensure desired behavior at train time + DG-enforcing optimization) is capable of ensuring the generalization of a particular property (accuracy parity) in practice while ensuring a good level of privacy. We answer this in the affirmative.
>
> For this rebuttal, however, we ran preliminary experiments comparing to DP-SGD-F of Xu et al. 2021 (see Section 6, paragraph “Disparate Impact of DP”). We did not manage to obtain good performance from DP-SGD-F in our setting on CelebA, possibly because the groups on CelebA are significantly more unbalanced than the datasets considered in their work. To proceed with the comparison, we evaluate the algorithms on the ADULT dataset that they used in their work. As subgroups, we consider four intersectional groups composed of all possible values of the “sex” attribute and prediction class.
>
> For a comparable epsilon value (0.69 for DP-SGD-F, and 0.7 for our DP-IS-SGD), we see that our method has smaller accuracy disparity (Eq. 2) across the groups. We show the results from the preliminary experiment on ADULT next:
>
> | Algorithm  | Accuracy disparity, p.p. | Overall accuracy |
> | ------------- |:-------------:|-------------:|
> | SGD                                        | 63 | **84**     |
> | DP-SGD                                  | 86 | 81     |
> | DP-SGD-F (Xu et al. 2021)     | 72 | 83     |
> | DP-IS-SGD (ours)                   | **31** | 75     |
>
> (Smaller accuracy disparity is better.)
>
> These results suggest that DP-IS-SGD, which is a simple application of our WYSIWYG framework, could perform better at disparity mitigation than methods from concurrent work.
>
> > Is DP necessary to get DG? Are there concrete examples where DG provable holds but the algorithm is not private?
>
> DP is strictly stronger than DG. In general it is possible to achieve provable DG in other ways, e.g., through compression (Bassily et al., 2016), or with a limited class of learners such as 1-NN (Nakkiran and Bansal, 2020).
>
> We focus on standard training with differential privacy for most of our experiments, because it is a rich area of research with multiple algorithms in the literature, that are also accessible via easy-to-use open-source frameworks such as opacus or diffprivlib. We show that any of this vast body of algorithms immediately implies strong guarantees of DG, and propose a framework on how to practically make use of this theoretical connection.
>
> > In general, it is not clear whether going the route of designing algorithms with DG property is the best way to get desired behavior (fairness, robustness, etc) in practice.
>
> DG is a formalization of the concept of any train-time behavior persisting at test time. Is the reviewer here referring to achieving DG through DP specifically? If so, we agree there could potentially be better ways to achieve DG than DP training. We explore this direction in our relaxation of Gaussian gradient noise regularizer.
>
> UPDATE: This response was edited to include the table and setup details for the experiment with DP-SGD-F.

---

### Author Response · Authors · 2022-08-02
**General response**

We thank all the reviewers for their time and high-quality reviews. We are glad that the reviewers found our WYSIWYG framework “interesting” ([cawT](https://openreview.net/forum?id=g05fHAvNeXx&noteId=naFUHFTXVTt), [4obB](https://openreview.net/forum?id=g05fHAvNeXx&noteId=EKnVtUeH21R), [QRKD](https://openreview.net/forum?id=g05fHAvNeXx&noteId=M2jaVdqaPJF)), “valuable” ([4obB](https://openreview.net/forum?id=g05fHAvNeXx&noteId=EKnVtUeH21R)), “experimentally justified” ([4obB](https://openreview.net/forum?id=g05fHAvNeXx&noteId=EKnVtUeH21R)), and “pointing to a promising direction for theory” ([XhyC](https://openreview.net/forum?id=g05fHAvNeXx&noteId=bLBUrOySXT4)). The reviews mentioned that our algorithms are “practical” and “make sense” ([XhyC](https://openreview.net/forum?id=g05fHAvNeXx&noteId=bLBUrOySXT4)), and found the experimental results “interesting” ([cawT](https://openreview.net/forum?id=g05fHAvNeXx&noteId=naFUHFTXVTt)), and “beautiful [as a] proof-of-concept” ([QRKD](https://openreview.net/forum?id=g05fHAvNeXx&noteId=M2jaVdqaPJF)). Finally, the reviews highlighted that the paper is “well-written” ([cawT](https://openreview.net/forum?id=g05fHAvNeXx&noteId=naFUHFTXVTt), [XhyC](https://openreview.net/forum?id=g05fHAvNeXx&noteId=bLBUrOySXT4)), the theory is “easy to understand” ([XhyC](https://openreview.net/forum?id=g05fHAvNeXx&noteId=bLBUrOySXT4)), and “expresses the concepts well” ([QRKD](https://openreview.net/forum?id=g05fHAvNeXx&noteId=M2jaVdqaPJF)).

We respond to the comments and questions of each review in individual comments.

---

### Author Response · Authors · 2022-08-09
**Looking for feedback/update from reviewers**

We thank you again for your thoughtful review and comments. With the author-reviewer period ending soon, we just wanted to reach out and see if any of the reviewers had any comments back to our rebuttal. We are looking for feedback on whether the points made in the reviews have now been addressed. We are happy to answer any remaining questions regarding our rebuttal or the paper itself. Thank you!

---

### Meta-Review · Area_Chair_DMDq · 2022-08-27

**Recommendation:** Accept
**Confidence:** Less certain

**Metareview:**

The consensus amongst the reviewers is that the connection between DP and DG, and the resulting WYSIWYG framework for providing generalization guarantees for suitably trained models is both of theoretical, and as the paper demonstrates, potentially practical interest. The paper does have some downsides, in particular, that the connection between DP and DG (and similar ideas, including information-theoretic bounds based on conditional mutual information) have been floating around in the literature, and these connections seem to have been somewhat missed by the authors. However, overall, based on the merit of the theoretical framework, and the interesting experimental results, I recommend that this paper be accepted.

**Award:**

No

---

### Decision · Program_Chairs · 2022-09-14

Accept